# Equivariant Maps for Hierarchical Structures

**Renhao Wang, Marjan Albooyeh**[*]
University of British Columbia,
{renhaow,albooyeh}@cs.ubc.ca

**Siamak Ravanbakhsh**
McGill University & Mila
siamak@cs.mcgill.ca

## Abstract

While using invariant and equivariant maps, it is possible to apply deep learning to a range of primitive data structures, a formalism for dealing with hierarchy is lacking. This is a significant issue because many practical structures are hierarchies of simple building blocks; some examples include sequences of sets, graphs of graphs, or multiresolution images. Observing that the symmetry of a hierarchical structure is the "wreath product" of symmetries of the building blocks, we express the equivariant map for the hierarchy using an intuitive combination of the equivariant linear layers of the building blocks. More generally, we show that *any* equivariant map for the hierarchy has this form. To demonstrate the effectiveness of this approach to model design, we consider its application in the semantic segmentation of point-cloud data. By voxelizing the point cloud, we impose a hierarchy of translation and permutation symmetries on the data and report state-of-the-art on SEMANTIC3D, S3DIS, and VKITTI, that include some of the largest real-world point-cloud benchmarks.

## 1 Introduction

In designing deep models for structured data, equivariance (invariance) of the model to transformation groups has proven to be a powerful inductive bias, which enables sample efficient learning. A widely used family of equivariant deep models constrain the feed-forward layer so that specific transformations of the input lead to the corresponding transformations of the output. A canonical example is the convolution layer, in which the constrained MLP is equivariant to translation operations. Many recent works have extended this idea to design equivariant networks for more exotic structures such as sets, exchangeable tensors and graphs, as well as relational and geometric structures.

This paper considers a nested hierarchy of such structures, or more generally, any hierarchical composition of transformation symmetries. These hierarchies naturally appear in many settings: for example, the interaction between nodes in a social graph may be a sequence or a set of events. Or in diffusion tensor imaging of the brain, each subject may be modeled as a set of sequences, where each sequence is a fibre bundle in the brain. The application we consider in this paper models point clouds as 3D images, where each voxel is a set of points with coordinates relative to the center of that voxel.

To get an intuition for a hierarchy of symmetry transformations, consider the example of a sequence of sequences – *e.g.*, a text document can be viewed as a sequence of sentences, where each sentence is itself a sequence of words. Here, each inner sequence as well as the outer sequence is assumed to possess an "independent" translation symmetry. Contrast this with symmetries of an image (2D translation), where all inner sequences (say row pixels) translate together, so we have a total of two translations. This is the key difference between the *wreath product* of two translation groups (former) and their *direct product* (latter). It is the wreath product that often appears in nested structures. As is evident from this example, the wreath product results in a significantly larger set of transformations, and as we elaborate later, it provides a stronger inductive bias.

---

[*]Currently at Borealis AI. Work done while at UBC.

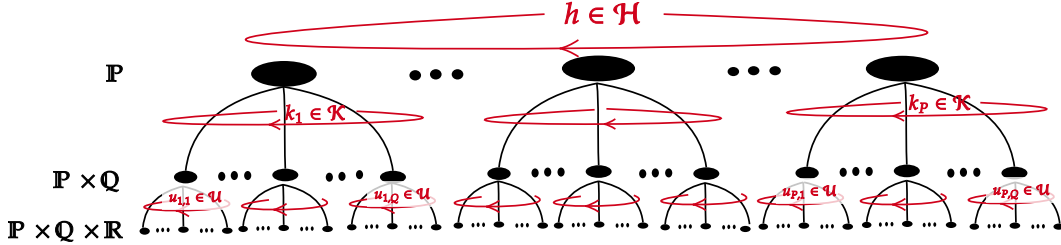

Figure 1: Wreath product can express the symmetries of hierarchical structures: wreath product of three groups $\mathcal{U} \wr \mathcal{K} \wr \mathcal{H}$ acting on the set of elements $\mathbb{P} \times \mathbb{Q} \times \mathbb{R}$, can be seen as *independent copies* of groups, $\mathcal{H}$, $\mathcal{K}$ and $\mathcal{U}$ at different level of hierarchy acting $\circlearrowleft$ on copies of $\mathbb{P}, \mathbb{Q}$ and $\mathbb{R}$. Intuitively, a linear map $\mathbf{W}_{\mathcal{U} \wr \mathcal{K} \wr \mathcal{H}} : \mathbb{R}^{PQR} \to \mathbb{R}^{PQR}$ equivariant to $\mathcal{U} \wr \mathcal{K} \wr \mathcal{H}$, performs pooling over leaves under each inner node ●, applies equivariant map for each inner structure (*i.e.*, $\mathbf{W}_{\mathcal{U}}$, $\mathbf{W}_{\mathcal{K}}$ and $\mathbf{W}_{\mathcal{H}}$ respectively), and broadcasting the output back to the leaves. A $\mathcal{U} \wr \mathcal{K} \wr \mathcal{H}$-equivariant map can be constructed as the sum of these three contributions, from equivariant maps at each level of hierarchy.

We are interested in application of equivariant/invariant deep learning to this type of nested structure. The building blocks of equivariant and invariant MLPs are *equivariant linear maps* of the feedforward layer. We show that any equivariant linear map for the hierarchical structure is built using equivariant maps for the individual symmetry group at different levels of the hierarchy. Our construction only uses additional pooling and broadcasting operations; see Fig. 1.

In the following, after discussing related works in Section 2, we give a short background on equivariant MLPs in Section 3. Section 4 starts by giving the closed form of equivariant maps for direct product of groups before moving to the more difficult case of wreath product in Section 4.2. Finally, Section 5 applies this idea to impose a hierarchical structure on 3D point clouds. We show that the equivariant map for this hierarchical structure achieves state-of-the-art performance on the largest benchmark datasets for 3D semantic segmentation.

## 2   Related Works

Group theory has a long history in signal processing [21], where in particular Fourier transformation and group convolution for Abelian groups have found tremendous success over the past decades. However, among non-commutative groups, wreath product constructions have been the subject of few works. Rockmore [37] give efficient procedures for fast Fourier transforms for wreath products. In a series of related works Foote et al. [14], Mirchandani et al. [32] investigate the wreath product for multi-resolution signal processing. The focus of their work is on the wreath product of cyclic groups for compression and filtering of image data.

Group theory has also found many applications in machine learning [24], and in particular deep learning. Design of invariant MLPs for general groups goes back to Shawe-Taylor [38]. More recently, several works investigate the design of equivariant networks for general finite [36] and infinite groups [7, 10, 25]. In particular, use of the wreath product for design of networks equivariant to a hierarchy of symmetries is briefly discussed in [36]. Equivariant networks have found many applications in learning on various structures, from image and text [29], to sets [34, 46], exchangeable matrices [20], graphs [1, 27, 30], and relational data [17], to signals on spheres [9, 26]. A large body of works investigate equivariance to Euclidean isometries; *e.g.*, [11, 40, 42].

When it comes to equivariant models for compositional structures, contributions have been sparse, with most theoretical work focusing on semidirect product (or more generally using induced representations from a subgroup) to model tensor-fields on homogeneous spaces [8, 10]. Direct product of symmetric groups have been used to model interactions across sets of entities [20] and in generalizations to relational data [17].

**Relation to Maron et al. [31]**   Maron et al. [31] extensively study equivariant networks for direct product of groups; in contrast we focus on wreath product. Since both of these symmetry transformations operate on the Cartesian product of $\mathcal{G}$-sets (see Fig. 1), the corresponding permutation groups are comparable. Indeed, direct product action is a sub-group of the imprimitive action of the

wreath product. The implication is that if sub-structures in the hierarchy transform independently (*e.g.*, in sequence of sequences), then using the equivariant maps proposed in this paper gives a strictly stronger inductive bias. However, if sub-structures move together (*e.g.*, in an image), then using a model equivariant to direct product of groups is preferable. One could also compare the learning bias of these models by noting that when using wreath product, the number of independent linear operators grows with the "sum" of independent operators for individual building blocks; in contrast, this number grows with the "product" of independent operators on blocks when using direct product. In the following we specifically contrast the use of direct product with wreath product for compositional structures.

# 3   Preliminaries

## 3.1   Group Action

A group $\mathcal{G} = \{g\}$ is a set equipped with a binary operation, such that the set is closed under this operation, $gh \in \mathcal{G}$, the operation is associative, $g(h\hbar) = (gh)\hbar$, there exists identity element $e \in \mathcal{G}$, and a unique inverse for each $g \in \mathcal{G}$ satisfying $gg^{-1} = e$. The action of $\mathcal{G}$ on a finite set $\mathbb{N}$ is a function $\alpha : \mathcal{G} \times \mathbb{N} \to \mathbb{N}$ that transforms the elements of $\mathbb{N}$, for each choice of $g \in \mathcal{G}$; for short we write $g \cdot n$ instead of $\alpha(g, n)$. Group actions preserve the group structure, meaning that the transformation associated with the identity element is identity $e \cdot n = n$, and composition of two actions is equal to the action of the composition of group elements $(gh) \cdot n = g \cdot (h \cdot n)$. Such a set, with a $\mathcal{G}$-action defined on it is called a $\mathcal{G}$-set. The group action on $\mathbb{N}$ naturally extends to $\mathbf{x} \in \mathbb{R}^{|\mathbb{N}|}$, where it defines a permutation of indices $g \cdot (x_1, \ldots, x_N) \doteq (x_{g \cdot 1}, \ldots, x_{g \cdot N})$. We often use a permutation matrix $\mathbf{G}^{(g)} \in \{0, 1\}^{N \times N}$ to represent this action – that is $\mathbf{G}^{(g)}\mathbf{x} = g \cdot \mathbf{x}$.

## 3.2   Equivariant Multilayer Perceptrons

A function $\phi : \mathbb{R}^{\mathbb{N}} \to \mathbb{R}^{\mathbb{M}}$ is equivariant to a given actions of group $\mathcal{G}$ iff $\phi(\mathbf{G}^{(g)}\mathbf{x}) = \tilde{\mathbf{G}}^{(g)}\phi(\mathbf{x})$ for any $\mathbf{x} \in \mathbb{R}^N$ and $g \in \mathcal{G}$. That is, a symmetry transformation of the input results in the corresponding symmetry transformation of the output. Note that the action on the input and output may in general be different. In particular, when $\tilde{\mathbf{G}}^{(g)} = \mathbf{I}_{\mathbb{M}}$ for all $g$ – that is, the action on the output is trivial – equivariance reduces to invariance. Here, $\mathbf{I}_{\mathbb{M}}$ is the $M \times M$ identity matrix. For simplicity and motivated by practical design choices, we assume the same action on the input and output.

For a feedforward layer $\phi : \mathbf{x} \mapsto \sigma(\mathbf{W}\mathbf{x})$, where $\sigma$ is a point-wise non-linearity and $\mathbf{W} \in \mathbb{R}^{N \times N}$, the equivariance condition above simplifies to commutativity condition $\mathbf{G}^{(g)}\mathbf{W} = \mathbf{W}\mathbf{G}^{(g)} \forall g \in \mathcal{G}$. This imposes a symmetry on $\mathbf{W}$ in the form of parameter-sharing [36, 43]. While we can use computational means to solve this equation for any finite group, an efficient implementation requires a closed form solution. Several recent works derive the closed form solutions for interesting groups and structures. Note that in general the feedforward layer may have multiple input and output channels, with identical $\mathcal{G}$-action on each channel. This only require replicating the parameter-sharing pattern in $\mathbf{W}$, for each combination of input and output channel. An Equivariant MLP is a stack of equivariant feed-forward layers, where the composition of equivariant layers is also equivariant. Therefore, our task in building MLPs equivariant to finite group actions is reduced to finding equivariant linear maps in the form of parameter-sharing matrices satisfying $\mathbf{G}^{(g)}\mathbf{W} = \mathbf{W}\mathbf{G}^{(g)} \forall g \in \mathcal{G}$; see Fig. 4(a.1,a.2) for $\mathbf{W}$ that are equivariant to $\mathcal{C}_4$, the group of circular translations of 4 objects (a.1), and symmetric group $\mathcal{S}_4$, the group of all permutations of 4 objects (a.2).

# 4   Equivariant Map for Product Groups

In this section we formalize the *imprimitive action* of the wreath product, which is used in describing the symmetries of hierarchical structures. We then introduce the closed form of linear maps equivariant to the wreath product of two groups. With hierarchies of more than two levels, one only needs to iterate this construction. To put this approach in perspective and to make the distinction clear, first we present the simpler case of direct product; see [31] for an extensive discussion of this setting.

## 4.1 Equivariant Linear Maps for Direct Product of Groups

The easiest way to combine two groups is through their direct product $\mathcal{G} = \mathcal{H} \times \mathcal{K}$. Here, the underlying set is the Cartesian product of the input sets and group operation is $(\hbar, \Bbbk)(\hbar', \Bbbk') \doteq (\hbar\hbar', \Bbbk\Bbbk')$. If $\mathbb{P}$ is an $\mathcal{H}$-set and $\mathbb{Q}$ a $\mathcal{K}$-set then the group $\mathcal{G} = \mathcal{H} \times \mathcal{K}$ naturally acts on $\mathbb{N} = \mathbb{P} \times \mathbb{Q}$ using $(\hbar, \Bbbk) \cdot (p, q) \doteq (\hbar \cdot p, \Bbbk \cdot q)$; see Fig. 2. This type of product is useful in modeling the Cartesian product of structures. The following claim characterises the equivariant map for direct product of two groups using the equivariant map for building blocks.

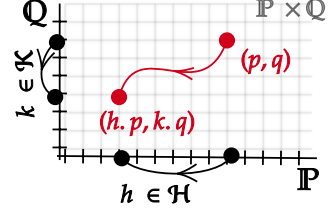

Figure 2: Direct product action.

> **Claim 1.** *Let $\mathbf{G}^{(\hbar)}$ represent $\mathcal{H}$-action, and let $\mathbf{W}_{\mathcal{H}} \in \mathbb{R}^{P \times P}$ be an equivariant linear map for this action. Similarly, let $\mathbf{W}_{\mathcal{K}} : \mathbb{R}^{Q \times Q}$ be equivariant to $\mathcal{K}$-action given by $\mathbf{G}^{(\Bbbk)}$ for $\Bbbk \in \mathcal{K}$. Then, the product group $\mathcal{G} = \mathcal{H} \times \mathcal{K}$ naturally acts on $\mathbb{R}^N = \mathbb{R}^{PQ}$ using $\mathbf{G}^{(g)} = \mathbf{G}^{(\hbar)} \otimes \mathbf{G}^{(\Bbbk)}$, and the Kronecker product $\mathbf{W}_{\mathcal{G}} = \mathbf{W}_{\mathcal{H}} \otimes \mathbf{W}_{\mathcal{K}}$, is a $\mathcal{G}$-equivariant linear map.*

Note that the claim is not restricted to permutation action.[2] The proof follows from the *mixed-product property* of the Kronecker product[3], and the equivariance of $\mathbf{W}_{\mathcal{H}}$ and $\mathbf{W}_{\mathcal{K}}$:

$$\mathbf{G}^{(g)} \mathbf{W}_{\mathcal{G}} = (\mathbf{G}^{(\hbar)} \otimes \mathbf{G}^{(\Bbbk)})(\mathbf{W}_{\mathcal{H}} \otimes \mathbf{W}_{\mathcal{K}}) = (\mathbf{G}^{(\hbar)} \mathbf{W}_{\mathcal{H}}) \otimes (\mathbf{G}^{(\Bbbk)} \mathbf{W}_{\mathcal{K}})$$
$$= (\mathbf{W}_{\mathcal{H}} \mathbf{G}^{(\hbar)}) \otimes (\mathbf{W}_{\mathcal{K}} \mathbf{G}^{(\Bbbk)}) = (\mathbf{W}_{\mathcal{H}} \otimes \mathbf{W}_{\mathcal{K}})(\mathbf{G}^{(\hbar)} \otimes \mathbf{G}^{(\Bbbk)}) = \mathbf{W}_{\mathcal{G}} \mathbf{G}^{(g)} \quad \forall g \in \mathcal{G}.$$

An implication of the tensor product form of $\mathbf{W}_{\mathcal{H} \times \mathcal{K}}$ is that the number of independent linear operators of the product map (free parameters in the parameter-sharing) is the product of the independent operators of the building blocks. Note that whenever dealing with product of inputs belonging to $\mathbb{R}^P$ and $\mathbb{R}^Q$, the result which belongs to $\mathbb{R}^{PQ}$ and therefore it is vectorized.

**Example 1** (Convolution in Higher Dimensions). *D-dimensional convolution is a Kronecker (tensor) product of one-dimensional convolutions. The number of parameters grows with the product of kernel width across all dimensions; Fig. 4(a.1) shows the parameter-sharing for circular 1D convolution $\mathbf{W}_{\mathcal{C}_4}$, and (b.1) shows the parameter-sharing for the direct product $\mathbf{W}_{\mathcal{C}_3 \times \mathcal{C}_4}$.*

**Example 2** (Exchangeable Tensors). *Hartford et al. [20] introduce a layer for modeling interactions across multiple sets of entities,* e.g.*, a user-movie rating matrix. Their model can be derived as the Kronecker product of 2-parameter equivariant layer for sets [46]. The number of parameters is therefore $2^D$ for a rank $D$ exchangeable tensor. Fig. 4(b.1) shows the 2-parameter model $\mathbf{W}_{\S_4}$ for sets, and (c.1) shows the parameter-sharing for the direct product $\mathbf{W}_{\mathcal{S}_3 \times \mathcal{S}_4}$.*

## 4.2 Wreath Product Action and Equivariance to a Hierarchy of Symmetries

Let us start with an informal definition. Suppose as before $\mathbb{P}$ and $\mathbb{Q}$ are respectively an $\mathcal{H}$-set and a $\mathcal{K}$-set. We can attach one copy of $\mathbb{Q}$ to each element of $\mathbb{P}$. Each of these inner sets or fibers have their own copy of $\mathcal{K}$ acting on them. Action of $\mathcal{H}$ on $\mathbb{P}$ simply permutes these fibers. Therefore the combination of all $\mathcal{K}$ actions on all inner sets combined with $\mathcal{H}$-action on the outer set defines the action of the wreath product on $\mathbb{P} \times \mathbb{Q}$. Fig. 3 demonstrates how one point $(p, q)$ moves under this action. Next few paragraphs formalize this.

**Semidirect Product.** Formally, wreath product is defined using *semidirect* product which is a generalization of direct product. In part due to its use in building networks equivariant to Euclidean isometries, application of semidirect product in building equivariant networks is explored in several recent works; see [10] and citations therein. In semidirect product, the underlying set (of group members) is again the product set. However the group operation is more involved. The (external) semi-direct product $\mathcal{G} = \mathcal{K} \rtimes_\gamma \mathcal{H}$, requires a *homomorphism* $\gamma : \mathcal{H} \to \mathrm{Aut}(\mathcal{K})$ that for each choice of

equivariant blocks

equivariant maps for product (×) and hierarchical (≀) structures

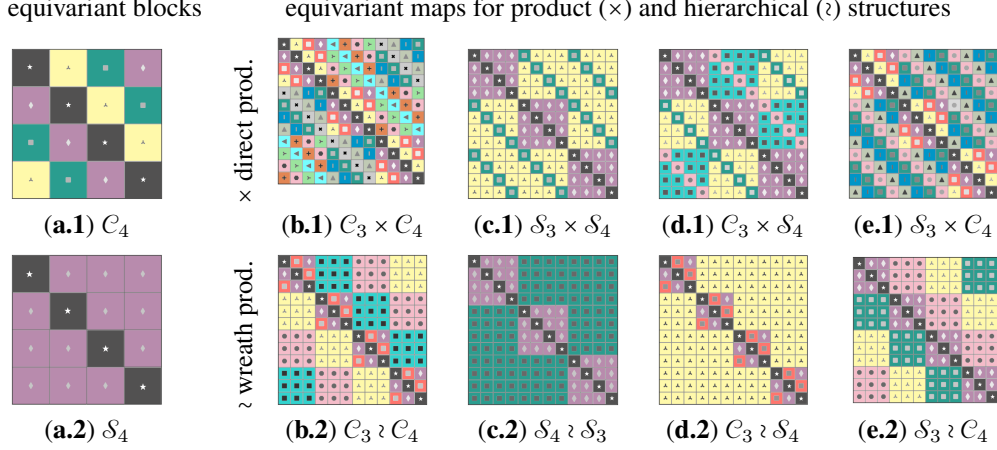

× direct prod.

≀ wreath prod.

**(a.1)** $\mathcal{C}_4$    **(b.1)** $\mathcal{C}_3 \times \mathcal{C}_4$    **(c.1)** $\mathcal{S}_3 \times \mathcal{S}_4$    **(d.1)** $\mathcal{C}_3 \times \mathcal{S}_4$    **(e.1)** $\mathcal{S}_3 \times \mathcal{C}_4$

**(a.2)** $\mathcal{S}_4$    **(b.2)** $\mathcal{C}_3 \wr \mathcal{C}_4$    **(c.2)** $\mathcal{S}_4 \wr \mathcal{S}_3$    **(d.2)** $\mathcal{C}_3 \wr \mathcal{S}_4$    **(e.2)** $\mathcal{S}_3 \wr \mathcal{C}_4$

Figure 4: Parameter-sharing patter in the equivariant linear maps for compositional structures. Equivariant maps for sequence and set are given in the first column (**a.1, a.2**). The first row shows the maps equivariant to various *products* of sets and sequences, while the second row is the corresponding map for *nested structure*. **Product Structures:** (**b.1**) product of sequences (as in image); (**c.1**) product of sets (exchangeable matrices); (**d.1**) product of a set and a sequence; (**e.1**) product of a sequence and a set. In the notation, the outer structure appears first in $\mathcal{H} \times \mathcal{K}$. **Hierarchical Structures:** (**b.2**) sequence of sequences (*e.g.*, multi-resolution sequence model); (**c.2**) nested set of sets; (**d.2**) set of sequences; (**e.2**) sequences of sets (similar to the model used in the application section.) In wreath product notation, the outer structure appears *second* in $\mathcal{K} \wr \mathcal{H}$. Note how the equivariant map for the hierarchy is a scaled version of the map for the outer structure with copies of the inner structure maps appearing as diagonal blocks.

$\hbar \in \mathcal{H}$, re-labels the elements of $\mathcal{K}$ while preserving its group structure. Using $\gamma$, the binary operation for the product group $\mathcal{G} = \mathcal{K} \rtimes_\gamma \mathcal{H}$ is defined as $(\hbar, \ell)(\hbar', \ell') = (\hbar\hbar', \ell \gamma_\hbar(\ell'))$. A canonical example is the semidirect product of translations ($\mathcal{K}$) and rotations ($\mathcal{H}$), which identifies the group of all rigid motions in the Euclidean space. Here, each rotation defines an automorphism of translations (*e.g.*, moving north becomes moving east after 90° clockwise rotation).

Now we are ready to define the wreath product of two groups. As before, let $\mathcal{H}$ and $\mathcal{K}$ denote two finite groups, and let $\mathbb{P}$ be an $\mathcal{H}$-set. Define $\mathcal{B}$ as the direct product of $P = |\mathbb{P}|$ copies of $\mathcal{K}$, and index these copies by $p \in \mathbb{P}$: $\mathcal{B} = \mathcal{K}_1 \times \ldots \times \mathcal{K}_p \times \ldots \times \mathcal{K}_P$. Each member of this group is a tuple $\ell = (\ell_1, \ldots, \ell_p, \ldots \ell_P)$. Since $\mathbb{P}$ is an $\mathcal{H}$-set, $\mathcal{H}$ also naturally acts on $\mathcal{B}$ by permuting the *fibers* $\mathcal{K}_p$. The semidirect product $\mathcal{B} \rtimes \mathcal{H}$ defined using this automorphism of $\mathcal{B}$ is called the wreath product, and written as $\mathcal{K} \wr \mathcal{H}$. Each member of the product group can be identified by the pair $(\hbar, \ell)$, where $\ell$ as a member of the *base* group itself is a $P$-tuple. This shows that the order of the wreath product group is $|\mathcal{K}|^P|\mathcal{H}|$ which can be much larger than the direct product group $\mathcal{K} \times \mathcal{H}$, whose order is $|\mathcal{K}||\mathcal{H}|$.

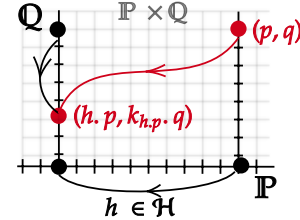

Figure 3: Imprimitive action of wreath product.

#### 4.2.1 Imprimitive Action of Wreath Product

If in addition to $\mathbb{P}$ being an $\mathcal{H}$-set, $\mathcal{K}$ action on $\mathbb{Q}$ is also defined, the wreath product group acts on $\mathbb{P} \times \mathbb{Q}$ (making it comparable to the direct product). Specifically, $(\hbar, \ell_1, \ldots, \ell_P) \in \mathcal{K} \wr \mathcal{H}$ acts on $(p, q) \in \mathbb{P} \times \mathbb{Q}$ as follows:

$$(\hbar, \ell_1, \ldots, \ell_P) \cdot (p, q) \doteq (\hbar \cdot p, \ell_{\hbar \cdot p} \cdot q).$$

Intuitively, $\mathcal{H}$ permutes the copies of $\mathcal{K}$ acting on each $\mathbb{Q}$, and itself acts on $\mathbb{P}$. We can think of $\mathbb{P}$ as the outer structure and $\mathbb{Q}$ as the inner structure; see Fig. 3.

**Example 3** (Sequence of Sequences). *Consider our early example where both $\mathcal{H} = \mathcal{C}_P$ and $\mathcal{K} = \mathcal{C}_Q$ are cyclic groups with regular action on $\mathbb{P} \cong \mathcal{H}$, $\mathbb{Q} \cong \mathcal{K}$. Each member of the wreath product $\mathcal{C}_Q \wr \mathcal{C}_P$, acts by translating each inner sequence using some $c \in \mathcal{C}_Q$, while $c' \in \mathcal{C}_P$ translates the outer sequence by $c'$.*

Let the $P \times P$ permutation matrix $\mathbf{G}^{(h)}$ represent the action of $h \in \mathcal{H}$, and the $Q \times Q$ matrices $\mathbf{G}^{(h_1)}, \dots, \mathbf{G}^{(h_P)}$ represent the action of $h_p$ on $\mathbb{Q}$. Then the action of $g \in \mathcal{K} \wr \mathcal{H}$ on (the vectorized) $\mathbb{P} \times \mathbb{Q}$ is the following $PQ \times PQ$ permutation matrix:

$$
\mathbf{G}^{(g)} = \begin{bmatrix} \mathbf{G}^{(h)}_{1,1}\mathbf{G}^{(h_1)}, & \dots, & \mathbf{G}^{(h)}_{1,P}\mathbf{G}^{(h_1)} \\ \vdots & \ddots & \vdots \\ \mathbf{G}^{(h)}_{P,1}\mathbf{G}^{(h_P)}, & \dots, & \mathbf{G}^{(h)}_{P,P}\mathbf{G}^{(h_P)} \end{bmatrix} = \sum_{p=1}^{P} \mathbf{1}_{p,h \cdot p} \otimes \mathbf{G}^{(h_p)} \tag{1}
$$

where $\mathbf{1}_{p,h \cdot p}$ is a $P \times P$ matrix whose only nonzero element is at row $p$ and column $h \cdot p$ with the value of 1; and $\mathbf{G}^{(h)}_{p,p'}$ is the element at row $p$ and column $p'$ of the permutation matrix $\mathbf{G}^{(h)}$. In the summation formulation, the Kronecker product $\mathbf{1}_{p,h \cdot p} \otimes \mathbf{G}^{(h_p)}$ puts a copy of permutation matrix $\mathbf{G}^{(h_p)}$ as a block of $\mathbf{G}^{(g)}$. Note that the resulting permutation matrix is different from the Kronecker product; in this case we have $P + 1$ (permutation) matrices participating in creating $\mathbf{G}^{(g)}$, compared with two matrices in the vanilla Kronecker product; see Fig. 5.

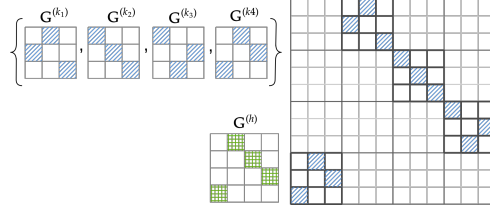

Figure 5: Permutation for the imprimitive wreath product ($\mathbf{G}^{(g)}$) on the right, is built from permutations of the outer structure using $\mathbf{G}^{(h)}$ and independent permutations of inner structures $\mathbf{G}^{(h_1)}, \dots, \mathbf{G}^{(h_4)}$.

#### 4.2.2 Equivariant Map

Consider a hierarchical structure, potentially with more than two levels of hierarchy, such as a set of sequences of images. Moreover, suppose that we have an equivariant map for each individual structure. The question answered by the following theorem is: how to use the equivariant map for each level to construct the equivariant map for the entire hierarchy? For now we only consider two levels of hierarchy; extension to more levels follows naturally, and is discussed later. Note that one instance of the hierarchy (*e.g.*, a sequence of sets) belongs to $\mathbb{R}^{PQ}$, because each instance of the structure is in $\mathbb{R}^{Q}$ and the outer structure is of size $P$ – that is vectorization of the input follows the hierarchy.

**Theorem 4.1.** *Let* $\mathbf{W}_{\mathcal{K} \wr \mathcal{H}} \in \mathbb{R}^{PQ \times PQ}$ *be the matrix of some linear map equivariant to the imprimitive action of* $\mathcal{K} \wr \mathcal{H}$ *on* $\mathbb{P} \times \mathbb{Q}$. *Any such map can be written as follows*

$$
\mathbf{W}_{\mathcal{K} \wr \mathcal{H}} = \mathbf{W}_{\mathcal{H}} \otimes \left( \mathbf{1}_Q \mathbf{1}_Q^\top \right) + \mathbf{I}_P \otimes \mathbf{W}_{\mathcal{K}}, \quad \text{where} \quad \mathbf{1}_Q = \overbrace{\left[ 1, \dots, 1 \right]^\top}^{Q \text{ times}}, \tag{2}
$$

$\mathbf{I}_P$ *is the* $P \times P$ *identity matrix,* $\mathbf{W}_{\mathcal{H}} \in \mathbb{R}^{P \times P}$ *is* $\mathcal{H}$-*equivariant, and* $\mathbf{W}_{\mathcal{K}} \in \mathbb{R}^{Q \times Q}$ *is equivariant to* $\mathcal{K}$-*action.*

Proof is in Appendix A. Pictorially, the first term of Eq. (2) scales up the matrix $\mathbf{W}_{\mathcal{H}}$, while the second term adds copies of $\mathbf{W}_{\mathcal{K}}$ on the diagonal blocks of the scaled matrix. Assuming *sets* of independent equivariant linear operators $\{\mathbf{W}_{\mathcal{K}}\}$ and $\{\mathbf{W}_{\mathcal{H}}\}$, from Eq. (2) it is evident that the *number of independent linear operators* equivariant to $\mathcal{K} \wr \mathcal{H}$ grows with the sum of those of the building blocks.[4] These operators may be combined using any parameter to create a parameterized linear map, which in turn can be expressed using parameter-sharing in a vanilla feed-forward layer. This in contrast with direct product in which the number of free parameters has a product form.

**Example 4** (Various Hierarchies of Sets and Sequences). *Consider two of the most widely used equivariant maps, translation equivariant convolution* $\mathbf{W}_{\mathcal{C}_P}$, *and* $\mathcal{S}_{\mathbb{Q}}$-*equivariant map which has the form* $\mathbf{W}_{\mathcal{S}_Q} = w_1 \mathbf{I}_Q + w_2$ *[46]. There are four combinations of these structures in a two level hierarchical structure: 1) set of sets* $\mathcal{S}_Q \wr \mathcal{S}_P$; *2) sequence of sequences* $\mathcal{C}_Q \wr \mathcal{C}_P$; *3) set of sequences* $\mathcal{C}_Q \wr \mathcal{S}_P$; *4) sequence of sets* $\mathcal{S}_Q \wr \mathcal{C}_P$. *Fig. 4(b.2-e.2) show the parameter-sharing matrix for these hierarchical structures, assuming a full kernel in 1D circular convolution.*

*The reader is invited to contrast the three parameter layer for set of sets, with the four parameter layer for interactions across sets in Fig. 4(c.1,c.2). Similarly, a model for sequence of sequences has far fewer parameters than a model for an image, as seen in Fig. 4(b.1, b.2).*

### 4.2.3 Deeper Hierarchies and Combinations with Direct Product

With more than two levels, the symmetry group involves more than one wreath product, which means that the equivariant map for the hierarchy is given by a recursive application of Theorem 4.1. For example, the equivariant map for $(\mathcal{K} \wr \mathcal{H}) \wr \mathcal{U}$, in which $\mathcal{U}$ acts on some set $\mathbb{R}$, and $\mathbf{W}_{\mathcal{U}}$ is the corresponding equivariant map, is given by $\mathbf{W}_{(\mathcal{K} \wr \mathcal{H}) \wr \mathcal{U}} = \mathbf{W}_{\mathcal{U}} \otimes (\mathbf{1}_{PQ} \mathbf{1}_{PQ}^{\top}) + \mathbf{I}_R \otimes \mathbf{W}_{\mathcal{K} \wr \mathcal{H}}$, where $\mathbf{W}_{\mathcal{K} \wr \mathcal{H}}$ is in turn given by Eq. (2). Note that wreath product is associative, and so the iterative construction above leads the same equivariant map as $\mathbf{W}_{\mathcal{K} \wr (\mathcal{H} \wr \mathcal{U})}$.

We can also mix and match this construction with that of direct product in Claim 1; for example, to produce the map for exchangeable tensors (product of sets), where each interaction is in the form of an image (hierarchy) – *i.e.*, the group is $(\mathcal{C}_R \times \mathcal{C}_V) \wr (\mathcal{S}_P \times \mathcal{S}_Q)$.

### 4.3 Efficient Implementation

With equivariant maps for direct product of groups, efficient implementation of $\mathbf{W}_{\mathcal{K} \times \mathcal{H}}$ using efficient implementation for individual blocks $\mathbf{W}_{\mathcal{K}}$, and $\mathbf{W}_{\mathcal{H}}$ is non-trivial – *e.g.*, consider 2D convolution. In contrast, it is possible to use a black-box implementation of the parameter-sharing layers for $\mathbf{W}_{\mathcal{K}}$, and $\mathbf{W}_{\mathcal{H}}$ to construct $\mathbf{W}_{\mathcal{K} \wr \mathcal{H}}$. To this end, let $\mathbf{x} \in \mathbb{R}^{PQ}$ be the input signal, and $\mathrm{mat}(\mathbf{x}) \in \mathbb{R}^{P \times Q}$ denote its matrix form; for example in a set of sets, each column is an inner set in this matrix form. Throughout, we are assuming a single input-output channel, relying on the idea that having multiple channels simply corresponds to replicating the linear map for each input-output channel combination. Then we can rewrite Eq. (2) as

$$\mathbf{W}_{\mathcal{K} \wr \mathcal{H}} \mathbf{x} = \mathrm{vec} \left( (\mathbf{W}_{\mathcal{H}} (\mathrm{mat}(\mathbf{x}) \mathbf{1}_Q)) \mathbf{1}_Q^{\top} + \mathrm{mat}(\mathbf{x}) \mathbf{W}_{\mathcal{K}} \right), \tag{3}$$

where the first multiplication $\mathrm{mat}(\mathbf{x}) \mathbf{1}_Q$ pools over columns (inner structures), and after application of the $\mathcal{H}$-equivariant map $\mathbf{W}_{\mathcal{H}}$ to the pooled value, the result is broadcasted back using right-multiplication by $\mathbf{1}_Q^{\top}$. The second term simply transforms each inner structure using $\mathbf{W}_{\mathcal{K}}$. The overall operation turns out to be simple and intuitive: the inner equivariant map is applied to individual inner structures, and the outer equivariant map is applied to pooled values and broadcasted back.

**Example 5** (Equivariant Map for Multiresolution Image). *Consider a coarse pixelization of an image into small patches. Eq. (3) gives the following recipe for a layer equivariant to independent translations within each patch as well as global translation of the coarse image: 1) apply convolution to each patch independently; 2) pool over each patch, apply convolution to the coarse image, and broadcast back to individual pixels in each patch. 3) add the contribution of these two operations. Notice how pooling over regions, a widely used operation in image processing, appears naturally in this approach. One could also easily extend this to more levels of hierarchy for larger images.*

## 5   Application: Point-Cloud Segmentation

In this section we consider a simple application of the theory above to large-scale point-cloud segmentation. The layer is the 3D version of equivariant linear map for sequence of sets, which is visualized in Fig. 4(d.2) – that is we combine translation and permutation symmetry by voxelizing the point-cloud; see Fig. 6. Using a hierarchical structure is beneficial compared to both the set model and 3D convolution. In particular, the set model lacks any prior of the Euclidean nature of the data, while the 3D convolution, in order to preserve resolution requires a fine-grained voxelization where each point appears in one voxel. The reader may ask: "how is the wreath product used to characterize the setting when the number of points is changing per voxel?" While our derivation using wreath

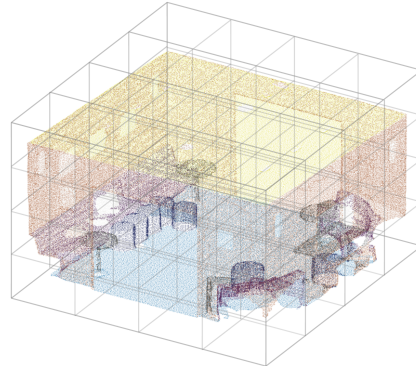

Figure 6: We impose a hierarchy of translation and permutation symmetry by voxelizing the point-cloud.

Table 1: Performance of various models on point cloud segmentation benchmarks.

| | SEMANTIC-8 | | S3DIS | | VKITTI | | |
|---|---|---|---|---|---|---|---|
| | **OA** | **mIoU** | **OA** | **mIoU** | **OA** | **mIoU** | **mAcc** |
| DEEPSETS[46] | 89.3 | 60.5 | 67.3 | 42.7 | 74.2 | 42.9 | 36.8 |
| POINTNET++[34] | 85.7 | 63.1 | 81.0 | 54.5 | 79.7 | 34.4 | 47.0 |
| SPG[28] | 92.9 | 76.2 | 85.5 | 62.1 | 84.3 | 67.3 | 52.0 |
| CONVPOINT[4] | 93.4 | 76.5 | 88.8 | 68.2 | - | - | - |
| KP-FCNN[39] | - | - | - | 70.6 | - | - | - |
| RSNET[22] | - | - | - | 56.5 | - | - | - |
| PCCN[41] | - | - | - | 58.3 | - | - | - |
| SNAPNET[5] | 91.0 | 67.4 | - | - | - | - | - |
| ENGELMANN ET AL. 2018[13] | - | - | - | - | 79.7 | 57.6 | 35.6 |
| ENGELMANN ET AL. 2017[12] | - | - | - | - | 80.6 | 54.1 | 36.2 |
| 3P-RNN[45] | - | - | - | - | 87.8 | 54.1 | 41.6 |
| WREATH PRODUCT NET. (ours) | 93.9 | 75.4 | 90.6 | 71.2 | 88.4 | 68.9 | 58.6 |
| WREATH PRODUCT NET. + ATTN (ours) | **95.2** | **77.4** | **95.8** | **80.1** | **90.7** | **69.5** | **59.2** |

product assumes a fixed number of points, since the resulting model for the set is independent of the number of data points, we can operate on various points per voxel. This is the same reason why we can apply the same convolution filter to images of different sizes.

The past few years have seen a growing body of work on learning with point cloud data; see [18] for a survey. Many methods use hierarchical aggregation and pooling; this includes the use of furthest point clustering for pooling in POINNET++[35], use of concentric spheres for pooling in SHELLNET, or KD-tree guided pooling in [23]. Several works extend convolution operation to maintain both translation and permutation equivariance in one way or another [3, 4, 41]; see also [6, 16]. Here, objective is not to introduce a radically new procedure, but to show the effectiveness of the approach discussed in previous sections in deep model design from first principles. Indeed, we are able to achieve state-of-the-art in several benchmarks for large point-clouds.

## 5.1 Equivariance to a Hierarchy of Translations and Permutations

Let $\mathbf{X} \in \mathbb{R}^{N \times C}$ be the input point cloud, where $N$ is the number of points and $C$ is the number of input channels (for concreteness, in this section are including the channel dimension in our formulae.) In addition to 3D coordinates, these channels may include RGB values, normal vectors or any other auxiliary information. Consider a voxelization of the point cloud with a resolution $D$ voxels per dimension, and consider the hierarchy of translation symmetry across voxels and permutation symmetry within each voxel. We may also replace 3D coordinates with relative coordinates within each voxel. Let $\Pi \in \{0, 1\}^{D^3 \times N}$ with one non-zero per column identify the voxel membership. Then the combination of equivariant set layer $\mathbf{X}\mathbf{W}_1 + \mathbf{1}\mathbf{1}^\top\mathbf{X}\mathbf{W}_2$ [46] with 3D convolution using the pool-broadcast interpretation given in Eq. (3), results in the following wreath-product equivariant linear layer

$$\phi(\mathbf{X}) = \mathbf{X}\mathbf{W}_1 + \Pi^\top\big(\mathbf{W}_3 * (\Pi\mathbf{X})\big), \tag{4}$$

where $\mathbf{W}_1 \in \mathbb{R}^{C \times C'}$, $*$ denotes the convolution operation, $\mathbf{W}_3 \in \mathbb{R}^{K^3 \times C \times C'}$ is the convolution kernel, with kernel width $K$, and $C'$ output channels. Here, multiplication with $\Pi$ and $\Pi^\top$ performs the pooling and broadcasting from/to points within voxels, respectively. Note that we have dropped the "set" operation $\Pi^\top((\Pi\mathbf{X})\mathbf{W}_2)$ that pools over each voxel, multiplies by the weight and broadcasts back to the points. This is because it can be absorbed in the convolution operation, and therefore it is redundant. In the equation above pooling operation can replace summation (implicit in matrix multiplication) with any other commutative operation; see [33, 44, 47], we use mean-pooling for the experiments. While the layer of Eq. (4) already achieves state-of-the-art in the experiments, we also consider adding an (equivariant) attention mechanism for further improvement; see Appendix B for details.

## 5.2 Empirical Results

We evaluate our model on two of the largest real world point cloud segmentation benchmarks, SE-MANTIC3D [19] and the Stanford Large-Scale 3D Indoor Spaces ( S3DIS) [2], as well as a dataset of virtual point cloud scenes, the VKITTI benchmark [15]. As shown in Table 1, in all cases we report new state-of-the-art. Table 2 compares the processing time and the size of different models. The architecture of WREATH PRODUCT NET. is a stack of equivariant layer Eq. (4) plus ReLU

Table 2: Comparison of pre-processing and training time and the number of parameters for SEMANTIC-8.

| Method | Pre-Proc. (hrs) | Train (hrs) | # Params. $\times 10^6$ |
|---|---|---|---|
| POINTNET | 8.82 | 3.54 | 3.50 |
| POINTNET++ | 8.84 | 7.46 | 12.40 |
| SNAPNET | 13.42 | 53.44 | 30.76 |
| SPG | 17.43 | 1.50 | 0.25 |
| CONVPOINT | 13.42 | 48.74 | 2.76 |
| OURS | 4.39 | 53.76 | 5.27 |
| OURS + ATTN | 4.39 | 91.68 | 47.01 |

nonlinearity and residual connections. WREATH PRODUCT NET.+ATTN also adds the attention mechanism. [5] Details on architecture and training, as well as further analysis of our results appear in Appendix C.

**Outdoor Scene Segmentation - SEMANTIC3D** [19] is the largest LiDAR benchmark dataset, consisting of 15 training point clouds and 15 tests point clouds with withheld labels, amassing altogether over 4 billion labeled points from a variety of urban and rural scenes. In particular, rather than working with the smaller REDUCED-8 variation, we run our experiments on the full dataset (SEMANTIC-8)[6]. Table 1 reports mIoU unweighted mean intersection over union metric (mIoU), as well as the overall accuracy (OA) for various methods. In Table 3 we report these measures for different voxelization resolution. At highest resolutions the performance degrades due to overfitting.

Table 3: Effect of voxelization resolution for SEMANTIC-8.

| # Voxels Per Dim | OA | mIoU |
|---|---|---|
| 2 | 81.7 | 62.7 |
| 3 | 85.9 | 67.3 |
| 4 | 90.6 | 70.5 |
| 5 | 92.3 | 72.2 |
| 6 | 93.7 | 72.8 |
| 7 | 94.4 | 74.1 |
| 8 | 95.1 | 75.6 |
| 9 | 94.6 | 77.1 |
| 10 | 95.2 | 77.4 |
| 12 | 94.8 | 73.0 |
| 16 | 90.6 | 71.5 |

**Indoor Scene Segmentation - Stanford Large-Scale 3D Indoor Spaces (S3DIS)** [2] consists of various 3D RGB point cloud scans from an assortment of room types on six different floor in three buildings on the Stanford campus, totaling almost 600 million points. Table 1 show that we achieve the best overall accuracy as well as mean intersection over union. This is in spite of the fact that our competition use extensive data-augmentation also for this dataset. In addition to random jittering and subsampling employed by both KPCONV and CONVPOINT, KPCONV also uses random dropout of RGB data.

**Virtual Scene Segmentation - Virtual KITTI (VKITTI)** [15] contains 35 monocular photo-realistic synthetic videos with fully annotated pixel-level labels for each frame and 13 semantic classes in total. Following [12], we project the 2D depth information within these synthetic frames into 3D space, thereby obtaining semantically annotated 3D point clouds. Note that VKITTI is significantly smaller than either SEMANTIC3D or S3DIS, containing only 15 millions points in total.

## Conclusion

This paper presents a procedure to design neural networks equivariant to hierarchical symmetries and nested structures. We describe how the imprimitive action of wreath product can formulate the symmetries of the hierarchy, contrast its use case with direct product, and characterize linear maps equivariant to the wreath product of groups. This analysis showed that we can always build an equivariant layer for the hierarchy using equivariant maps for the building blocks, and additional pool-and-broadcast operations. We consider one of the many use cases of this approach to design a deep model for large-scale semantic segmentation of point cloud data, where we are able to achieve state-of-the-art using a simple architecture.

## Broader Impact

As deep learning finds its way in various real-world applications, the practitioners are finding more constrains in representing their data in formats and structures amenable to existing deep architectures. The list of basic structures such as images, sets, and graphs that we can approach using deep models has been growing over the past few years. The theoretical contribution of this paper substantially expands this list by enabling deep learning on a hierarchy of structures. This could potentially unlock new applications in data-poor and structure-rich settings. The task we consider in our experiments is deep learning with large point-cloud data, which is finding growing applications, from autonomous vehicles to geographical surveys. While this is not a new task, our empirical results demonstrate the effectiveness of the proposed methodology in dealing with hierarchy in data structure.

## Acknowledgments and Disclosure of Funding

This research was in part supported by CIFAR AI Chairs program and NSERC Discovery Grant. Computational resources were provided by Mila and Compute Canada.

## Footnotes

[2]The tensor product of irreducible representation of two distinct finite groups is an irreducible representation for the product group. Therefore this construction of equivariant maps can be used with a decomposition into irreducible representations for the general linear case.

[3]$(\mathbf{A} \otimes \mathbf{B})(\mathbf{C} \otimes \mathbf{D}) = (\mathbf{AC}) \otimes (\mathbf{BD})$

[4]In the proof we show that the number of parameters is the sum of those of the building blocks minus one.

[5]We have released a PYTORCH implementation of our models at `https://github.com/rw435/wreathProdNet`

[6]The leader-board can be viewed at `http://www.semantic3d.net/view_results.php?chl=1`

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
