[Supplementary Material]

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

# A  Proof of Theorem 4.1

*Proof.* We show that both of the two terms in Eq. (2) are equivariant to $\mathcal{K} \wr \mathcal{H}$-action as expressed by the permutation matrix of Eq. (1).

**Part 1.**  We show that $\mathbf{G}^{(g)}$ commutes with the first term in Eq. (2) for any $g \in \mathcal{K} \wr \mathcal{H}$:

$$\left(\mathbf{W}_{\mathcal{H}} \otimes (\mathbf{1}_Q \mathbf{1}_Q^\top)\right) \mathbf{G}^{(g)} = \left(\mathbf{W}_{\mathcal{H}} \otimes (\mathbf{1}_Q \mathbf{1}_Q^\top)\right) \left(\sum_{p=1}^{P} \mathbf{1}_{p, \hbar \cdot p} \otimes \mathbf{G}^{(\hbar_p)}\right) \tag{5}$$

$$= \sum_{p=1}^{P} \left(\mathbf{W}_{\mathcal{H}} \mathbf{1}_{p, \hbar \cdot p}\right) \otimes \overbrace{\left((\mathbf{1}_Q \mathbf{1}_Q^\top) \mathbf{G}^{(\hbar_p)}\right)}^{=(\mathbf{1}_Q \mathbf{1}_Q^\top)} \tag{6}$$

$$= \mathbf{W}_{\mathcal{H}} \left(\sum_{p=1}^{P} \mathbf{1}_{p, \hbar \cdot p}\right) \otimes (\mathbf{1}_Q \mathbf{1}_Q^\top) \tag{7}$$

$$= \left(\mathbf{W}_{\mathcal{H}} \mathbf{G}^{(\hbar)}\right) \otimes (\mathbf{1}_Q \mathbf{1}_Q^\top) = \left(\mathbf{G}^{(\hbar)} \mathbf{W}_{\mathcal{H}}\right) \otimes (\mathbf{1}_Q \mathbf{1}_Q^\top) \tag{8}$$

$$= \sum_{p=1}^{P} \left(\mathbf{1}_{p, \hbar \cdot p} \mathbf{W}_{\mathcal{H}}\right) \otimes \left((\mathbf{1}_Q \mathbf{1}_Q^\top) \mathbf{G}^{(\hbar_p)}\right) \tag{9}$$

$$= \sum_{p=1}^{P} \left(\mathbf{1}_{p, \hbar \cdot p} \otimes \mathbf{G}^{(\hbar_p)}\right) \left(\mathbf{W}_{\mathcal{H}} \otimes (\mathbf{1}_Q \mathbf{1}_Q^\top)\right) \tag{10}$$

$$= \left(\sum_{p=1}^{P} \mathbf{1}_{p, \hbar \cdot p} \otimes \mathbf{G}^{(\hbar_p)}\right) \left(\mathbf{W}_{\mathcal{H}} \otimes (\mathbf{1}_Q \mathbf{1}_Q^\top)\right) = \mathbf{G}^{(g)} \left(\mathbf{W}_{\mathcal{H}} \otimes (\mathbf{1}_Q \mathbf{1}_Q^\top)\right). \tag{11}$$

In Eq. (5) we substitute from Eq. (1). In Eq. (6) we pulled the summation out, and then used the mixed product property of Kronecker product. We also note that a permutation of the uniformly constant matrix $\mathbf{1}_Q \mathbf{1}_Q^\top$ is constant. In Eq. (7), since the summation can be restricted to the first term we pull out $\mathbf{W}_{\mathcal{H}}$. In Eq. (8) we use the fact that the summation of the previous line is the permutation matrix $\mathbf{G}^{(\hbar)}$ and apply the key assumption that $\mathbf{W}_{\mathcal{H}}$ and $\mathbf{W}_{\mathcal{K}}$ are equivariant to $\mathcal{H}$ and $\mathcal{K}$-action respectively. The following lines repeat the procedure so far in reverse order to commute $\mathbf{G}^{(g)}$ and $\left(\mathbf{W}_{\mathcal{H}} \otimes (\mathbf{1}_Q \mathbf{1}_Q^\top)\right)$.

**Part 2.**  For the second term, again we use the mixed-product property and equivariance of the input maps

$$\left(\mathbf{I}_P \otimes \mathbf{W}_{\mathcal{K}}\right) \mathbf{G}^{(g)} = \left(\mathbf{I}_P \otimes \mathbf{W}_{\mathcal{K}}\right) \left(\sum_{p=1}^{P} \mathbf{1}_{p, \hbar \cdot p} \otimes \mathbf{G}^{(\hbar_p)}\right) \tag{12}$$

$$= \sum_{p=1}^{P} \left(\mathbf{I}_P \mathbf{1}_{p, \hbar \cdot p}\right) \otimes \left(\mathbf{W}_{\mathcal{K}} \mathbf{G}^{(\hbar_p)}\right) \tag{13}$$

$$= \sum_{p=1}^{P} \left(\mathbf{1}_{p, \hbar \cdot p} \mathbf{I}_P\right) \otimes \left(\mathbf{G}^{(\hbar_p)} \mathbf{W}_{\mathcal{K}}\right) \tag{14}$$

$$= \sum_{p=1}^{P} \left(\mathbf{1}_{p, \hbar \cdot p} \otimes \mathbf{G}^{(\hbar_p)}\right) \left(\mathbf{I}_P \otimes \mathbf{W}_{\mathcal{K}}\right) = \mathbf{G}^{(g)} \left(\mathbf{I}_P \otimes \mathbf{W}_{\mathcal{K}}\right) \tag{15}$$

where in Eq. (14) we used the fact that the identity matrix commutes with any matrix, as well as the equivariance of $\mathbf{G}^{(\hbar_p)}$. Eqs. (13) and (15) simply use the mixed product property. Putting the two parts together shows the equivariance of the first and second term in Eq. (2), completing the proof. $\square$

## A.1 Proof of Maximality

Previously we proved that the linear maps of Eq. (2) is equivariant to imprimitive action of wreath product $\mathcal{K} \wr \mathcal{H}$ given by Eq. (1). Here, we prove that any $\mathcal{K} \wr \mathcal{H}$ equivariant linear map has this form.

**Claim 2.** *Assuming that any $\mathcal{K}$-equivariant ($\mathcal{H}$-equivariant) linear map $\mathbf{W}_{\mathcal{K}}$ ($\mathbf{W}_{\mathcal{H}}$) can be written as a linear combination of $K$ ($H$) independent linear operators, then any $W_{\mathcal{K} \wr \mathcal{H}}$ as defined in Eq. (2) is a linear combination of $K + H - 1$ independent linear bases.*

*Proof.* Consider the two terms (I) $\mathbf{W}_{\mathcal{H}} \otimes (\mathbf{1}_Q \mathbf{1}_Q^\top)$ and (II) $\mathbf{I}_P \otimes \mathbf{W}_{\mathcal{K}}$ in Eq. (2), where (I) has $H$ independent bases and (II) has $K$ independent bases. If they had independent bases, their summation would have $K + H$ linear bases. However, (I) and (II) have exactly one shared basis: $\mathbf{W}_{\mathcal{H}} = \mathbf{I}_P$ is $\mathcal{H}$-equivariant, and $\mathbf{W}_{\mathcal{K}} = \mathbf{1}_Q \mathbf{1}_Q^\top$ is $\mathcal{K}$-equivariant. Therefore, $\mathbf{I}_P \otimes (\mathbf{1}_Q \mathbf{1}_Q^\top)$ is a basis for both (I) and (II). $\square$

Next, we show that any $\mathcal{K} \wr \mathcal{H}$-equivariant linear map can be written using as a weighted sum of $K + H - 1$ independent linear operators. Together with proof of equivariance of $\mathbf{W}_{\mathcal{K} \wr \mathcal{H}}$ these two facts prove that any $\mathcal{K} \wr \mathcal{H}$ equivariant linear map has the form of Eq. (2).

**Claim 3.** *Any linear map that is equivariant to the imprimitive action of $\mathcal{K} \wr \mathcal{H}$ can a linear combination of $K + H - 1$ equivariant linear bases.*

*Proof.* Let $\mathbf{W} \in \mathbb{R}^{N \times N}$ be the matrix representing this equivariant map, satisfying $\mathbf{W} \mathbf{G}^g = \mathbf{G}^g \mathbf{W}$ $\quad \forall g \in \mathcal{K} \wr \mathcal{H}$. Since $\mathbf{G}^{g\top} \mathbf{W} \mathbf{G}^g = \mathbf{W}$ $\quad \forall g \in \mathcal{K} \wr \mathcal{H}$, all elements of the matrix $\mathbf{W}$ that are in the same orbit are tied together. This constraint means that the number of unique values in $\mathbf{W}$ are equal to the number of orbits in this simultaneous action on rows and columns of $\mathbf{W}$. Using a property of Kronecker product, we can also write the equation above as $\text{vec}(\mathbf{G}^g \mathbf{W} \mathbf{G}^{g\top}) = (\mathbf{G}^g \otimes \mathbf{G}^g) \text{vec}(\mathbf{W})$. Therefore, the diagonal group action is given by $\mathbf{G}^g \otimes \mathbf{G}^g$. Using *Burnside's lemma* we get the number of orbits $L$ in this action to be

$$L = \frac{1}{|\mathcal{G}|} \sum_{g \in \mathcal{K} \wr \mathcal{H}} \text{Tr}(\mathbf{G}^{(g)} \otimes \mathbf{G}^{(g)}) = \sum_{g \in \mathcal{K} \wr \mathcal{H}} \text{Tr}(\mathbf{G}^{(g)})^2 \tag{16}$$

Our objective is to show that $L = H + K - 1$. Our strategy is to replace the definition of $\mathbf{G}^g$ from Eq. (1) into Eq. (16) and simplify. This simplification involves numerous steps.

$$L = \frac{1}{|\mathcal{H}||\mathcal{K}|^P} \sum_{\hbar \in \mathcal{H}, \hbar_1, \ldots, \hbar_P \in \mathcal{K}} \text{Tr}\left(\sum_{p=1}^{P} \mathbf{1}_{p, \hbar \cdot p} \otimes \mathbf{G}^{(\hbar_p)}\right)^2 \tag{17}$$

$$= \frac{1}{|\mathcal{H}||\mathcal{K}|^P} \sum_{\hbar \in \mathcal{H}, \hbar_1, \ldots, \hbar_P \in \mathcal{K}} \text{Tr}\left(\sum_{p=1}^{P} \mathbf{G}^{(\hbar)}_{p,p} \mathbf{G}^{(\hbar_p)}\right)^2 \tag{18}$$

$$= \frac{1}{|\mathcal{H}||\mathcal{K}|^P} \sum_{\hbar \in \mathcal{H}, \hbar_1, \ldots, \hbar_P \in \mathcal{K}} \left(\sum_{p=1}^{P} \mathbf{G}^{(\hbar)}_{p,p} \text{Tr}\left(\mathbf{G}^{(\hbar_p)}\right)\right)\left(\sum_{p'=1}^{P} \mathbf{G}^{(\hbar)}_{p',p'} \text{Tr}\left(\mathbf{G}^{(\hbar_{p'})}\right)\right) \tag{19}$$

$$= \frac{1}{|\mathcal{H}||\mathcal{K}|^P} \sum_{\hbar \in \mathcal{H}, \hbar_1, \ldots, \hbar_P \in \mathcal{K}} \left(\sum_{p=1}^{P} \mathbf{G}^{(\hbar)}_{p,p} \text{Tr}\left(\mathbf{G}^{(\hbar_p)}\right)^2\right) +$$

$$\left(\sum_{p=1}^{P} \sum_{p' \neq p, p'=1}^{P} \mathbf{G}^{(\hbar)}_{p,p} \mathbf{G}^{(\hbar)}_{p',p'} \text{Tr}\left(\mathbf{G}^{(\hbar_p)}\right) \text{Tr}\left(\mathbf{G}^{(\hbar_{p'})}\right)\right) \tag{20}$$

where in Eq. (18) we observed that the contribution from $\mathbf{G}^{(\hbar_p)}$ to the trace of $\mathcal{G}^{(g)}$ is non-zero only if $\mathbf{G}^{(\hbar)}_{p,p} = 1$. Note that all the $\mathbf{G}$ matrices are permutation matrices and in the following we will continue to use this fact without elaborating. In Eq. (19) we used linearity of trace, and in Eq. (20) we divided the product of the previous line into $p = p'$ and $p \neq p'$. As we show below, the first term simplifies to $K$ and the second terms simplifies to $H - 1$, showing that $L = K + H - 1$. We start by

simplifying the first term of Eq. (20) below:

$$\frac{1}{|\mathcal{H}||\mathcal{K}|^P} \sum_{\hbar \in \mathcal{H}, \hbar_1, \ldots, \hbar_P \in \mathcal{K}} \left( \sum_{p=1}^{P} \mathbf{G}_{p,p}^{(\hbar)} \operatorname{Tr}\left(\mathbf{G}^{(\hbar_p)}\right)^2 \right) \tag{21}$$

$$= \frac{1}{|\mathcal{H}||\mathcal{K}|^P} \sum_{p=1}^{P} \left( \sum_{\hbar \in \mathcal{H}} \mathbf{G}_{p,p}^{(\hbar)} \right) \left( \sum_{\hbar_1, \ldots, \hbar_P \in \mathcal{K}} \operatorname{Tr}(\mathbf{G}^{(\hbar_p)})^2 \right) \tag{22}$$

$$= \frac{1}{|\mathcal{H}||\mathcal{K}|^P} \left( \sum_{\hbar \in \mathcal{H}} \sum_{p=1}^{P} \mathbf{G}_{p,p}^{(\hbar)} \right) \left( |\mathcal{K}|^{P-1} \sum_{\hbar \in \mathcal{K}} \operatorname{Tr}(\mathbf{G}^{(\hbar)})^2 \right) \tag{23}$$

$$= \frac{1}{|\mathcal{H}||\mathcal{K}|^P} \left( \sum_{\hbar \in \mathcal{H}} \operatorname{Tr}(\mathbf{G}^{(\hbar)}) \right) \left( |\mathcal{K}|^{P-1} |\mathcal{K}| K \right) \tag{24}$$

$$= \frac{1}{|\mathcal{H}||\mathcal{K}|^P} |\mathcal{H}||\mathcal{K}|^P K = K \tag{25}$$

where beside simple algebra, in Eq. (24), we used the Burnside lemma on $\mathbf{W}_{\mathcal{K}}$, which gives us the number of unique parameters (orbits) as $K = \frac{1}{|\mathcal{K}|} \sum_{\hbar \in \mathcal{K}} \operatorname{Tr}(\mathbf{G}^{(\hbar)})^2$. Similarly for $\mathcal{H}$-action we get $\frac{1}{|\mathcal{H}|} \sum_{\hbar \in \mathcal{H}} \operatorname{Tr}(\mathbf{G}^{(\hbar)})^2 = H$. In Eq. (25) we used the fact that $\mathcal{H}$ action on $\mathbb{P}$ is transitive (*i.e.*, it has a single orbit), to get $|\mathcal{H}| = \sum_{\hbar \in \mathcal{H}} \operatorname{Tr}(\mathbf{G}^{(\hbar)})$. Similarly, due to transitivity of $\mathcal{K}$-action on $\mathbb{Q}$ we have $|\mathcal{K}| = \sum_{\hbar \in \mathcal{K}} \operatorname{Tr}(\mathbf{G}^{(\hbar)})$. Next, we use these same equalities to simplify the second term of Eq. (20):

$$\frac{1}{|\mathcal{H}||\mathcal{K}|^P} \sum_{\hbar \in \mathcal{H}, \hbar_1, \ldots, \hbar_P \in \mathcal{K}} \sum_{p=1}^{P} \sum_{p' \neq p, p'=1}^{P} \mathbf{G}_{p,p}^{(\hbar)} \mathbf{G}_{p',p'}^{(\hbar)} \operatorname{Tr}\left(\mathbf{G}^{(\hbar_p)}\right) \operatorname{Tr}\left(\mathbf{G}^{(\hbar_{p'})}\right) \tag{26}$$

$$= \frac{1}{|\mathcal{H}||\mathcal{K}|^P} \sum_{p=1}^{P} \sum_{p' \neq p, p'=1}^{P} \left( \sum_{\hbar \in \mathcal{H}} \mathbf{G}_{p,p}^{(\hbar)} \mathbf{G}_{p',p'}^{(\hbar)} \right) \left( |\mathcal{K}|^{P-2} \left( \sum_{\hbar \in \mathcal{K}} \operatorname{Tr}(\mathbf{G}^{(\hbar_{p'})}) \right)^2 \right) \tag{27}$$

$$= \frac{1}{|\mathcal{H}||\mathcal{K}|^P} \left( \sum_{\hbar \in \mathcal{H}} \left( \sum_{p=1}^{P} \sum_{p'=1}^{P} \mathbf{G}_{p,p}^{(\hbar)} \mathbf{G}_{p',p'}^{(\hbar)} \right) - \left( \sum_{p=1}^{P} \mathbf{G}_{p,p}^{(\hbar) 2} \right) \right) \left( |\mathcal{K}|^{P-2} |\mathcal{K}|^2 \right) \tag{28}$$

$$= \frac{1}{|\mathcal{H}|} \sum_{\hbar \in \mathcal{H}} \left( \sum_{p=1}^{P} \mathbf{G}_{p,p}^{(\hbar)} \right) \left( \sum_{p'=1}^{P} \mathbf{G}_{p',p'}^{(\hbar)} \right) - \left( \sum_{p=1}^{P} \mathbf{G}_{p,p}^{(\hbar)} \right) \tag{29}$$

$$= \frac{1}{|\mathcal{H}|} \sum_{\hbar \in \mathcal{H}} \operatorname{Tr}(\mathbf{G}^{(\hbar)})^2 - \operatorname{Tr}(\mathbf{G}^{(\hbar)}) = H - 1. \tag{30}$$

Eqs. (25) and (30) together with Eq. (16) show that $L = H + K - 1$, that any linear map $\mathbf{W}_{\mathcal{G}}$ equivariant to imprimitive action of $\mathcal{K} \wr \mathcal{H}$, can be written using $H + K - 1$ independent linear bases. Considering this proof together with our proof for the first part (that the equivariant map of Eq. (2) has $H + K - 1$ independent linear bases) completes the proof that any $\mathcal{K} \wr \mathcal{H}$-equivariant linear map is of the form Eq. (2). $\square$

## B  Adaptive Pooling and Attention

While the layer of Eq. (4) performs competitively, to improve its performance we add a novel attention mechanism. To this end we complement the pool-and-broadcast scheme of Eq. (4) with an adaptive pooling mechanism using a learned $\tilde{\Pi} : \mathbb{R}^N \to \Delta_L$ in Eq. (4), where $\Delta_L$ is the $L$-dimensional probability simplex. Here, one may interpret $L$ as the number of *latent classes*, analogous to $D^3$ voxels. The pooling matrix $\tilde{\Pi}$, is a function of input through a linear map $\mathbf{W}_3^{(\ell)} \in \mathbb{R}^{C \times L}$ followed by $\operatorname{Softmax} : \mathbb{R}^L \to (0,1)^L$, so that the probability of each point belonging to different classes sums to one – that is

$$\tilde{\Pi}_n = \operatorname{Softmax}\left( (\mathbf{X}\mathbf{W}_3)_n \right) \forall n \in [N]. \tag{31}$$

We then use a linear function that models the interaction between latent classes, for each pair of channels. This is done using a rank four tensor $\mathbf{W}_4 \in \mathbb{R}^{L \times L \times C \times C}$. As before the result is broadcasted

back using $\tilde{\Pi}_n^\top$, giving us the adaptive pooling layer, in which the output for channel $c$ is given by

$$\phi(\mathbf{X})_c = \sum_{c'=1}^{C'} \tilde{\Pi}^\top \mathbf{W}_{4,c,c'} \tilde{\Pi} \mathbf{X}_c. \tag{32}$$

Intuitively this layer can decide which subset of nodes to pool, and therefore acts like an equivariant attention mechanism. In experiments, we see further improvement in results when adding this nonlinear layer to the linear layer of Eq. (4).

## C  Details on Experiments

### C.1  Architecture and Data Processing

Our results are produced using minor architectural variations and limited hyperparameter search. Concretely, our best-performing models on SEMANTIC3D and S3DIS involve 56 residual-style blocks, where each block comprises of 2 equivariant linear maps of Eq. (4). Each map involves a single 3D convolution operator with periodic padding and a $3 \times 3 \times 3$ kernel. We use an identity map to facilitate skip connections within these blocks. The number channels is fixed 64, except the final block, which maps to the number of segmentation classes (8 for SEMANTIC3D and 13 for S3DIS). When incorporating adaptive pooling, we utilize 5, 10, 15, 25 and 50 latent classes $L$ for blocks 1-19, 19-29, 30-39, 40-49 and 50-56, respectively and inclusively. Our best-performing model on VKITTI uses 22 such residual-style blocks with identical architectural hyperparameters. For adaptive pooling, we then use 5, 10, 15, and 20 latent classes for blocks 1-9, 10-14, 15-19, and 20-22, respectively and inclusively.

Additionally, while previous works rely on stochastic point cloud subsampling during pre-processing and reprojection during post-processing, we simply split large point clouds into subsets of 1 million points each to ameliorate memory issues. We then compute a $9 \times 9 \times 9$ voxelization over each sample, and train on mini-batches of 4 such samples. To generate predictions, we stitch predictions on these smaller samples together.

### C.2  Datasets: Training, Validation and Test

**Outdoor Scene Segmentation - SEMANTIC3D SEMANTIC3D [20]**   We hold out 4 of the available 15 training point clouds to use as a validation set, as in [30].

Table 4 reports the overall accuracy (OA), and mean intersection over union (mIoU) for our method and the competition. We achieve both the best overall accuracy as well as the best mean intersectio over union.

| Method | OA | mIoU | Man-Made Terrain | Natural Vegetation | High Vegetation | Low Vegetation | Buildings | Hardscape | Scanning Artifacts | Cars |
|---|---|---|---|---|---|---|---|---|---|---|
| DEEPSETS[50] | 89.3 | 60.5 | 90.8 | 76.3 | 41.9 | 22.1 | 94.0 | 46.5 | 26.8 | 85.4 |
| POINTNET[36] | 85.7 | 63.1 | 81.9 | 78.1 | 64.3 | 51.7 | 75.9 | 36.4 | 43.7 | 72.6 |
| SNAPNET[5] | 91.0 | 67.4 | 89.6 | 79.5 | 74.8 | 56.1 | 90.9 | 36.5 | 34.3 | 77.2 |
| SPG[30] | 92.9 | 76.2 | 91.5 | 75.6 | **78.3** | 71.7 | 94.4 | **56.8** | 52.9 | 88.4 |
| CONVPOINT[4] | 93.4 | 76.5 | 92.1 | 80.6 | 76.0 | **71.9** | 95.6 | 47.3 | **61.1** | 87.7 |
| WREATH PRODUCT NET. (ours) | 93.9 | 75.4 | 93.4 | 84.0 | 76.4 | 68.2 | 96.8 | 45.6 | 47.4 | 91.9 |
| WREATH PRODUCT NET. + ATTN (ours) | **94.6** | **77.1** | **95.2** | **87.1** | 75.3 | 67.1 | **96.1** | 51.3 | 51.0 | **93.4** |

Table 4: Performance of various models on the *full* SEMANTIC3D dataset (semantic-8). Higher is better, bolded is best. mIoU is unweighted mean intersection over union metric. OA is overall accuracy. Per class splits show mIoU.

**Indoor Scene Segmentation - Stanford Large-Scale 3D Indoor Spaces (S3DIS)**   The S3DIS dataset [2] consists of various 3D RGB point cloud scans from an assortment of room types on six different floor in three buildings on the Stanford campus, totaling almost 600 million points. Following previous works by [12, 30, 36, 37, 42], we perform 6-fold cross validation with micro-averaging, computing all metrics once over the merged predictions of all test folds.

Here, we achieve the best overall accuracy as well as mean intersection over union. This is in spite of the fact that our competition use extensive data-augmentation also for this dataset. In particular, both KPCONV and CONVPOINT use random jittering and subsampling.

Figure 7: (**top left**) RGB scan of a point cloud with 280 994 028 points from the SEMANTIC3D dataset; (**middle left**) segmentation results from our method; (**bottom left**) ground truth segmentation; (**top right**) RGB scan of a point cloud with 19 767 991 points from the SEMANTIC3D dataset. (**middle right**) segmentation results from our method; (**bottom right**) ground truth segmentation;

| Method | OA | mIoU | Ceiling | Floor | Wall | Beam | Column | Window | Door | Chair | Table | Bookcase | Sofa | Board | Clutter |
|---|---|---|---|---|---|---|---|---|---|---|---|---|---|---|---|
| DEEPSETS[50] | 67.3 | 42.7 | 81.1 | 72.4 | 67.2 | 16.9 | 25.8 | 44.2 | 48.5 | 51.0 | 49.8 | 21.7 | 24.4 | 17.2 | 34.6 |
| POINTNET[36] | 78.5 | 47.6 | 88.0 | 88.7 | 69.3 | 42.4 | 23.1 | 47.5 | 51.6 | 42.0 | 54.1 | 38.2 | 9.6 | 29.4 | 35.2 |
| RSNET[23] | N/A | 56.5 | 92.5 | 92.8 | 78.6 | 32.8 | 34.4 | 51.6 | 68.1 | 60.1 | 59.7 | 50.2 | 16.4 | 44.9 | 52.0 |
| PCCN[45] | N/A | 58.3 | 92.3 | 96.2 | 75.9 | 0.27 | 6.0 | 69.5 | 63.5 | 65.6 | 66.9 | 68.9 | 47.3 | 59.1 | 46.2 |
| SPG[30] | 85.5 | 62.1 | 89.9 | 95.1 | 76.4 | 62.8 | 47.1 | 55.3 | 68.4 | 73.5 | 69.2 | 63.2 | 45.9 | 8.7 | 52.9 |
| CONVPOINT[4] | 88.8 | 68.2 | 95.0 | **97.3** | 81.7 | 47.1 | 34.6 | 63.2 | 73.2 | **75.3** | **71.8** | 64.9 | 59.2 | 57.6 | 65.0 |
| KP-FCNN[43] | N/A | 70.6 | 93.6 | 92.4 | 83.1 | 63.9 | 54.3 | 66.1 | 76.6 | 57.8 | 64.0 | 69.3 | **74.9** | 61.3 | 60.3 |
| WREATH PRODUCT NET. (ours) | 90.6 | 71.2 | 94.3 | 96.7 | 80.6 | 65.0 | 76.2 | 62.1 | 71.9 | 64.3 | 62.7 | 60.2 | 68.8 | 63.5 | 59.9 |
| WREATH PRODUCT NET. + ATTN (ours) | **95.8** | **80.1** | **97.2** | 97.8 | **88.6** | **72.3** | **82.0** | **73.6** | **78.6** | 75.1 | 72.0 | **73.1** | 71.4 | **82.1** | **77.2** |

Table 5: Performance of various models on the S3DIS dataset (micro-averaged over all 6 folds). Higher is better, bolded is best. mIoU is unweighted mean intersection over union metric. OA is overall accuracy. Per class splits show mIoU.

## C.3 Virtual Scene Segmentation - Virtual KITTI

The VKITTI dataset [16] contains 35 monocular photo-realistic synthetic videos with fully annotated pixel-level labels for each frame and 13 semantic classes in total. Following [12], we project the 2D depth information within these synthetic frames into 3D space, thereby obtaining semantically annotated 3D point clouds. Similar to the training and evaluation scheme in [12, 29], we separate the original set of sequences into 6 non-overlapping subsequences and use a 6 fold cross-validation protocol (with micro-averaging similar to the methodology on S3DIS).

Note that VKITTI is significantly smaller than either SEMANTIC3D or S3DIS, containing only 15 millions points in total. We hypothesize that these smaller, and sparser point clouds provide little geometric signal outside vegetation and road structure. This partially explains our only incremental improvement

| Method | OA | mIoU | mAcc |
|---|---|---|---|
| DEEPSETS[50] | 74.2 | 42.9 | 36.8 |
| POINTNET[36] | 79.7 | 34.4 | 47.0 |
| ENGELMANN ET AL. 2018[13] | 79.7 | 57.6 | 35.6 |
| ENGELMANN ET AL. 2017[12] | 80.6 | 54.1 | 36.2 |
| 3P-RNN[49] | 87.8 | 54.1 | 41.6 |
| SPG[29] | 84.3 | 67.3 | 52.0 |
| WREATH PRODUCT NET. (ours) | 88.4 | 68.9 | 58.6 |
| WREATH PRODUCT NET. + ATTN (ours) | **90.7** | **69.5** | **59.2** |

Table 6: Performance of various models on the VKITTI dataset (micro-averaged over all 6 folds). Higher is better, best results are in bold. mIoU is unweighted mean intersection over union metric. OA is overall accuracy. mAcc is mean accuracy.

over the state of the art (see Table 6). However, we expect future simulations of point cloud scenes to become increasingly dense, in line with increasingly powerful LiDAR scanners for real world applications [14, 40], where our simple baselines can potentially produce significant gain both in accuracy and computation.