[Reviews · NeurIPS 2020]

Review 1

Summary and Contributions: The paper tackles the problem of learning hierarchical structures (for example set of sequences, image of images ) by equivariant networks. To this end, the authors formulate the symmetry groups for these learning setups and inputs and characterize the maximally expressive linear equivariant layers that can serve as building blocks for deep invariant or equivariant networks. The authors demonstrate the applicability of their layers and networks on 3D point cloud recognition problems and achieve very nice results.

Strengths: * Important problem: the paper targets an important problem that can be useful for many tasks. * The characterization of equivariant layers for wreath product symmetries is elegant. * The application part is both convincing and suggests elegant modeling of point cloud analysis problems by a voxelization+permutation equivariance. This might be beneficial for people working in this field in the future.

Weaknesses: Overall I like the paper and I don’t find significant flaws.

Correctness: Seems OK.

Clarity: Yes.

Relation to Prior Work: Overall the paper is well positioned with respect to previous work.

Reproducibility: Yes

Additional Feedback: * Figure 1 caption: typo innder->inner. * Figure 4 confused me a lot. What are the assumptions of the vectorization of the * input vector? Is c.2 correct? It would be beneficial to revisit this figure and make sure that everything is correct and understandable. * L 170 - missing reference. * Example 4 - can the authors provide algebraic formulations for these examples by applying equation 2? * L 248 “effectiveness of approach” missing “the”. * I think that there is a similar characterization of equivariant layers in the general direct product setup in "On learning sets of Symmetric Elements" ICML 2020. Can the authors comment on that? * A lot of related works discuss the approximation power of equivariant MLPs. Can the authors comment/dicuss this aspect as well? ------------ Post author response: I read the author response as well as the other reviews. I like the paper a lot and will keep my score.


Review 2

Summary and Contributions: This work considers the problem of constructing linear maps that are equivariant to the action of groups with hierarchical structure, building on a line of recent work on developing neural network layers that are equivariant by construction. The approach developed in the paper is based on the concept of wreath products of groups; by representing these products in matrix form, the paper demonstrates that these products can be efficiently implemented in practice. The experimental section discusses an application to point cloud segmentation, where a hierarchy of translation equivariance across voxels and permutation equivariance within each voxel is desired.

Strengths: 1. The paper develops a sound theoretical foundation for the notion of hierarchical equivariance by identifying its relationship with the wreath product of groups. A simple, efficiently implementable representation of this class of operators is cleanly derived from this formalism. The derivation also yields as a byproduct an interesting connection to the commonly-used pooling operator. 2. The presentation is clear and easy to follow throughout, aided by the inclusion of helpful examples and figures. 3. The work reports strong empirical performance on 3D point cloud segmentation benchmarks. This is an encouraging validation of the merits of enforcing equivariance by construction within the architecture of the network. The proposed adaptive pooling mechanism may also be of independent interest to practitioners in the field.

Weaknesses: 1. While the reported empirical results are strong, it is not clear from the text how the wreath product nets used in the experiments compare to the baselines in terms of model complexity (say, in terms of number of free parameters as a rough guide). In particular, it seems that the 4th order interactions introduced in the attention layers could greatly inflate the number of parameters in the model. Could the baselines be made stronger simply through the use of a larger network?

Correctness: I did not identify any issues with correctness.

Clarity: The paper is very clearly written. I appreciated the inclusion of examples throughout Section 4 -- these were helpful in grounding the concepts discussed in the text. While Figure 4 was a useful reference, it should be made more explicit that shapes are used to denote parameter sharing; the figure could perhaps also be made more easily interpretable through the use of color.

Relation to Prior Work: I was satisfied with the coverage of prior work in the paper.

Reproducibility: Yes

Additional Feedback: Minor comments: - There is a broken reference in L170. --- After author response: I am satisfied with the author response and will maintain my acceptance recommendation.


Review 3

Summary and Contributions: The authors are interested in studying symmetries for data with a hierarchical structure. In particular, they wish to build networks that respect those structures in a practical and efficient way. The paper starts with some theoretical discussions (centred around the concept of 'wreath product of groups') on hierarchical data with symmetries at every level, before proposing some neural networks architectures that are applied to pointcloud datasets. Finally, the experiment sections looks at a hierarchy of symmetries, with translational symmetry at the level of the voxelisation of a pointcloud, and permuation symmetry within each voxel.

Strengths: There is a fast growing literature that shows that networks that respect symmetries by construction are more efficient than those that don't (even if trained to respect the symmetry via data-augmentation). The authors add to this literature with some very convincing architecture proposal. The subject is nicely introduced, despite its difficulty, and the experiments are convincing.

Weaknesses: I think section would benefit from some additional explanations. The most important explanation missing is, in my opinion, what is the exact wreath group that is being used. The authors do mention that the group is built from translations of the voxel, and from permuations of points within each voxel, but there is something unclear here: the number of points within each voxel will vary from voxel to voxel, so what is the exact permutation group that is being used, and how does it act with varying number of points. Another missing point is a brief discussion on how to choose the number of voxels D along each dimension. This choice will have various impact that need to be taken into account: for example, increasing D will lead to a more fine grained account of translation symmetry, but will also decrease the account of permutation symmetry. The value of D will also affect the size of the wreath group, and therefore the amount of compute and memory.

Correctness: I could not find anything that appeared wrong.

Clarity: Overall, yes. But please, do see my comments above about section 5.

Relation to Prior Work: Yes

Reproducibility: Yes

Additional Feedback: * I could not find the definition of the group C_k before example 3, even though that group is used multiple times. Please move the definition of that group earlier (at the same time you define the symmetric group for example). * Broken citation on line 170 ============================= Update after Authors feedback ============================= Thank you for your feedback. This is indeed an excellent paper which I thoroughly enjoyed reading, and learnt a lot from. Congratulations.


Review 4

Summary and Contributions: The paper proposes formalism of weight-tying in neural networks to achieve desirable equivariance. Compositions and hierarchies of equivariances and symmetries are analysed to design wreath-product based operation for weight-tying of kernels in a neural network. A neural network is designed to be equivariant to a hierarchy of translations and permutations for the task of point cloud segmentation. The the proposed network achieves SOTA results on various benchmarks.

Strengths: - Code in the supplementary - Novel theoretical grounding for extending various equivariances to neural networks - Strong empirical evaluation - Well written paper

Weaknesses: - Evaluation is only on one task, where one hierarchy of equivariances is evaluated. The paper could've been more impressive with multiple examples of hierarchies applied to different tasks.

Correctness: The claims and method in the paper seem correct, I did not check the proof in the appendix.

Clarity: The paper is generally well written.

Relation to Prior Work: Some related work on 3D equivariance and weight-tying should be cited "Cubenet: Equivariance to 3d rotation and translation" by Worrall et al, and "Learning to convolve: A generalized weight-tying approach" by Diaconu et al.

Reproducibility: Yes

Additional Feedback: - L36 and L43: I'm not sure I agree with "stronger inductive bias", it seems that wreath product "provides more flexibility". - "innder" in caption of Figure 1. - "patter" in caption of Figure 4. - I don't think the diagrams in Figure 4 are very clear. Adding the patterns for C_3, S_3 would help, so that it is clear what are the inputs to direct and wreath product with C_4 and S_4. Coloring the cells could also help. - Figure 4: The white cell in the diagonal of the patterns is also a shared weight, so maybe also add a symbol to the white cells? I suspect that the diagram was drawn with the assumption that the white cell is identity or 1, otherwise the symbols on the diagonal blocks in b-e should also change? - L140: I don't think using "fibers" without any definition helps with informal explanation of wreath product. Consider simplifying the informal definition even further, with a concrete example. I liked the translation and rotation example in L156-159. Figure 2 and 3 could also be much easier to understand when there are definitions for H and K that are intuitive and familiar. - L170: broken reference/citation. - L182: Is it P+1 permutation matrices or P permutation matrices? The summation in Eq (1) summing P terms, each containing one permutation matrix, right? Or are you referring to the "sum" of 1_{p, h*p} matrices as 1 additional permutation matrix? - In Example 5, L229-235, the translation in each patch needs to be cyclic (i.e. pixels translated outside the patch are wrapped around and put on the opposite end of the patch) in order to be fully equivariant. Is that right? --- Thank you for the replies, I'm looking forward to updated diagrams and informal explanation.

[Author Response · NeurIPS 2020]

We thank all the reviewers for their thorough reading of the paper, and we are happy to see their positive feedback. A common question was regarding Figure 4 and its example. We will apply suggested edits to make it more clear. Below we address other questions and comments.

**Reviewer 1.**

- Q: What are the assumptions of the vectorization of the input vector?
  - In the vectorization, the inner structure is repeated within the outer structure. We will revisit Figure 4 as suggested.
- Q: there is a similar characterization of equivariant layers in the general direct product setup in [Maron et al'20], Can the authors comment on that?
  - This is a very relevant paper. Both models are valid and each can be used in a different setting. In contrast to Maron et al'20 we note that hierarchical structures often have wreath product symmetry (i.e., substructures "move" independently) and focus on this type of group action. We plan to further discuss that paper.
- Q. Can the authors comment/discuss [the approximation power] as well?
  - We now have a proof of *maximality*: the proposed linear map of Eq (2) is the most expressive equivariant linear map for the given action, assuming input linear maps ($\mathbf{W}_{\mathcal{K}}$ and $\mathbf{W}_{\mathcal{H}}$) are also maximal. This will be added to the paper. The question of universality remains open.

**Reviewer 2.**

| Method | Pre-Proc. (hrs) | Train (hrs) | # Params. $\times 10^6$ |
|---|---|---|---|
| POINTNET | 8.82 | 3.54 | 3.50 |
| POINTNET++ | 8.84 | 7.46 | 12.40 |
| SNAPNET | 13.42 | 53.44 | 30.76 |
| SPG | 17.43 | 1.50 | 0.25 |
| CONVPOINT | 13.42 | 48.74 | 2.76 |
| OURS | 4.39 | 53.76 | 5.27 |
| OURS + ATTN | 4.39 | 91.68 | 47.01 |

- Q: compare to the baselines in terms of model complexity (say, in terms of number of free parameters as a rough guide). In particular, it seems that the 4th order interactions introduced in the attention layers could greatly inflate the number of parameters in the model?
  - The following table compares the preprocessing and training time, as well as the number of parameters of our model and the competition for SEMANTIC3D dataset. Using attention indeed significantly increases the number of parameters. Note that we achieve SOTA even without using attention. We will add this table and more discussions on efficiency to the revised version.

| # Voxels Per Dim | Train (hrs) | Accuracy | mean IoU |
|---|---|---|---|
| 2 | 42.60 | 81.7 | 62.7 |
| 3 | 48.77 | 85.9 | 67.3 |
| 4 | 56.12 | 90.6 | 70.5 |

**Reviewer 3.**

- Q: [...] how is the wreath product used when the number of points are changing per voxel (rephrased)
  - This is a theoretically valid concern and we will clarify the text to avoid confusion. Since the number of parameters of the equivariant set layer does not change with the size of the set, we can have different number of points per voxel. The same logic allows DeepSets to be applied to point-clouds of different size, or a convolution filter to be applied to images of different size and so on.
- Q: [...] a brief discussion on how to choose the number of voxels D along each dimension. This choice will have various impact [...]
  - The reviewer is correct: increasing $D$ increases the accuracy as well as the training time. The results in the following table shows this trend for coarser voxelizations (due to limited time). However, assuming all voxels remain (non-empty), changing $D$ does not affect the number of parameters and the size of activations (which is proportional to the number of points), and so its effect on memory usage is minimal. We will add an extended version of this table to the paper.

**Reviewer 4.**

- Q: About wreath product imposing a "stronger inductive bias" compared to direct product.
  - Direct product "action" is in fact a subgroup of the imprimitive wreath product "action" as a permutation group. This means that their equivariant maps are directly comparable, and wreath product produces a more constrained layer. We will add this argument to support our statement.
- Q: Is it P+1 permutation matrices or P permutation matrices?
  - There are $P$ permutations, one for each inner structure (blocks) and 1 permutation for the outer structure.
- Q: [...] the translation in each patch needs to be cyclic in order to be fully equivariant. Is that right?
  - Yes. However, while it is customary to work with cyclic groups, one could "assume" patches of input are zero padded by half the kernel width. This makes the theory applicable while having no effect in the actual implementation without cyclic assumption.

[Meta-Review · NeurIPS 2020]

The paper attempts to improve invariance/equivariance modelling in neural network. Specifically, authors target the problem of nested symmetries or invariances. For example, in point cloud of a scene there can be permutation equivariance at object level as well as point level (set of sets). Many other nesting are important (sets of rotations/translations, image of images, sets of voxels etc.) and so the paper is solving an important problem that can be useful for many tasks. The paper develops a sound theoretical foundation for the such nested/hierarchical equivariance by identifying its relationship with the wreath product of groups. A simple, efficiently implementable representation of this class of operators is cleanly derived from this formalism. The derivation also yields as a byproduct an interesting connection to the commonly-used pooling operator. All the reviewers found the characterization of equivariant layers using wreath product symmetries to be novel and elegant. Finally, the paper shows strong empirical performance gains on standard point cloud analysis problems by imposing set of voxel equivariance (i.e. voxelization+permutation). Overall, all the reviewers appreciated the paper a lot and thus I am happy to recommend an acceptance to NeurIPS. For camera ready version please add proof of maximality as well as timing/efficiency comparison. In future, authors should take better precautions in maintaining privacy when providing leaderboard links.